# Kono-S Anastomosis in Crohn’s Disease: A Retrospective Study on Postoperative Morbidity and Disease Recurrence in Comparison to the Conventional Side-To-Side Anastomosis

**DOI:** 10.3390/jcm11236915

**Published:** 2022-11-23

**Authors:** Matthias Kelm, Joachim Reibetanz, Mia Kim, Kathrin Schoettker, Markus Brand, Alexander Meining, Christoph-Thomas Germer, Sven Flemming

**Affiliations:** 1Department of General, Visceral, Transplantation, Vascular and Pediatric Surgery, Center of Operative Medicine (ZOM), University Hospital of Wuerzburg, 97080 Wuerzburg, Germany; 2Department of Colorectal Surgery, Munich Hospital, Hospital Neuperlach/Harlaching, 81737 Munich, Germany; 3Division of Gastroenterology, Department of Internal Medicine II, University Hospital of Wuerzburg, 97080 Wuerzburg, Germany

**Keywords:** Crohn’s disease, surgical therapy, ileocecal resection, Kono-S anastomosis

## Abstract

Introduction: The rates of postoperative recurrence following ileocecal resection due to Crohn’s disease remain highly relevant. Despite this fact, while the Kono-S anastomosis technique initially demonstrated promising results, robust evidence is still lacking. This study aimed to analyze the short- and long-term outcomes of the Kono-S versus side-to-side anastomosis. Methods: A retrospective single-center study was performed including all patients who received an ileocecal resection between 1 January 2019 and 31 December 2021 at the Department of Surgery at the University Hospital of Wuerzburg. Patients who underwent conventional a side-to-side anastomosis were compared to those who received a Kono-S anastomosis. The short- and long-term outcomes were analyzed for all patients. Results: Here, 29 patients who underwent a conventional side-to-side anastomosis and 22 patients who underwent a Kono-S anastomosis were included. No differences were observed regarding short-term postoperative outcomes. The disease recurrence rate postoperatively was numerically lower following the Kono-S anastomosis (median Rutgeert score of 1.7 versus 2.5), with a relevantly increased rate of patients in remission (17.2% versus 31.8%); however, neither of these results reached statistical significance. Conclusion: The Kono-S anastomosis method is safe and feasible and potentially decreases the severity of postoperative disease remission.

## 1. Introduction

Crohn’s disease (CD) represents a major socioeconomic burden and challenges health care systems worldwide. Despite the great advances made in medical treatment, including the introduction of biologics, CD remains incurable and the rates of surgery remain high. The current evidence shows that many patients with CD need surgery at least once during their lifetime [1,2].

Previously, the indications for surgery have focused on CD-related complications such as fistulas or stenosis. However, the current studies and updated guidelines consider ileocecal resection (ICR) in cases of localized terminal ileitis already at early time points [3,4]. This adaptation is mainly based on data from the LIR!C trial by Ponsioen et al., which demonstrated an improved quality of life and decreased need for medical treatment after ICR in patients with localized terminal ileitis compared to patients receiving infliximab [5]. These positive results were not only confirmed in the short- but also long-term follow-up and were underlined by other studies that demonstrated similar results in terms of quality of life and medical therapy [6,7]. However, while the advantage for laparoscopic ICR has been clearly shown in many studies in comparison to open surgery, the optimal technique for the creation of the anastomosis is still controversial [8,9,10]. Despite the significant progress made on CD therapy by demonstrating the potential of early ICR, with its implementation in the current guidelines, the rates of postoperative recurrence remain a major issue for many patients. Many theories suggest that the anastomosis has a central role in postoperative disease recurrence [11,12]. Thus, the perfect technique for the anastomosis has been an ongoing matter of debate [13,14]. To overcome this issue, Kono et al. introduced a novel technique (Kono-S anastomosis), which is based on the idea that the inflammation in CD originates from the mesentery. In line with this, Kono et al. proposed that the anastomosis should be created away from the mesentery (an anti-mesenteric handsewn anastomosis). The initial limited data from his group demonstrated impressive results in terms of postoperative recurrence [15,16]. However, further evidence about the technical feasibility of the Kono-S anastomosis, including the short-term morbidity and postoperative recurrence rates, especially during implementation and in a non-selective cohort, is lacking [17]. As such, there is currently no clear recommendation for a technique to create the anastomosis following ICR. Therefore, the goal of our study was to investigate the feasibility and potential of the Kono-S anastomosis in comparison with the conventional side-to-side anastomosis in patients who received ICR due to ileitis terminalis in a non-selective cohort.

## 2. Materials and Methods

### 2.1. Study Population

In this single-center retrospective study, all patients with ileocecal resection (ICR) due to Crohn’s disease who were operated on between 1 January 2019 and 31 December 2021 at the Department of Surgery at the University Hospital of Wuerzburg were evaluated. The included patients were suffering from terminal inflammatory (Montreal classification L1 and L3), penetrating, or stricturing ileitis (Montreal classification B1–B3), while the patients who needed extended resection or strictureplasty, had received a diversion, or had a history of ulcerative colitis were excluded. Preoperatively, the extent of inflammation was assessed via endoscopy and an MRI scan. The indication for an operation was discussed by a multidisciplinary team, including a gastroenterologist, IBD surgeon, and radiologist.

All patients were divided into two subgroups depending on the type of anastomosis. After the introduction of the Kono-S anastomosis at our hospital in 2021, almost all patients received the handsewn Kono-S technique, while all patients who were operated on before bowel reconstruction received the conventional (anisoperistaltic) side-to-side stapler anastomosis. By using those distribution criteria, an evaluation of the safety and feasibility of the Kono-S anastomotis during a potential learning curve was possible. The patients were usually operated on laparoscopically with extracorporal creation of the anastomosis. The Kono-S anastomosis was performed as described by Kono as a functional end-to-end handsewn anastomosis [15]. For the analysis, sociodemographic and clinicopathological data, including on the time of diagnosis, history of the disease, as well as the immunosuppressive or anti-inflammatory medication history, were collected for each patient from their patient records. In addition, an evaluation of the preoperative disease extent and a postoperative histopathological analysis of the resected specimen were performed and included the level of inflammation (mucosal damage, immune cell inflammation), as well as the extent at the resection margins (positive resection margins for inflammation).

### 2.2. Outcome

The primary endpoint was the Rutgeert score in the first endoscopy postoperatively, usually after 6–12 months. The Rutgeert score is captured endoscopically and represents an established method to define the extent of inflammation at the anastomosis site following ICR. The secondary endpoints included the rates of conversion and surgical and non-surgical postoperative complications, as well as the length of hospital stay.

### 2.3. Statistical Analysis

The descriptive data are presented as medians with the range or total numbers with the percentages. The differences in patient characteristics were assessed using a *t*-test, Fisher’s exact test, or ANOVA test in accordance with the data scale and distribution. A *p*-value of <0.05 was considered statistically significant. In addition, the effect size was included by calculating the value of Cohen’s *d*, with values > 0.45 being considered a relevant effect size. The statistical analysis was done using GraphPad Prism (Version 8.0.0 for Windows, GraphPad Software, San Diego, CA, USA).

### 2.4. Ethical Approval

Ethical approval for this study was obtained from the Ethics Committee of the University of Wuerzburg, Germany.

## 3. Results

### 3.1. Patient Cohort

In this retrospective single-center study, 78 patients received ileocecal resection at the Department of Surgery at the University Hospital of Wuerzburg between 2019 and 2021. Of those, 27 patients were excluded from the study due to loss during follow-up, participation in a study, or delayed postoperative endoscopy. Therefore, 51 patients were finally included in the study (Figure 1). Twenty-nine patients from our cohort received a conventional side-to-side stapler anastomosis, whereas 22 patients were reconstructed with the functional end-to-end, handsewn Kono-S anastomosis. As presented in Table 1, the groups did not differ in terms of patient characteristics such as age, BMI, co-morbidities, or smoking habits, or in their preoperative levels of hemoglobin and albumin. In addition, both groups were comparable regarding their disease history, including for previous CD-associated surgery (*n* = 6/group, *p* = 0.59), whereas the rates of preoperative immunosuppressive medication trended higher for patients who received the side-to-side anastomosis (21 versus 10, *p* = 0.05). However, the rates of postoperative immunosuppression were comparable between both groups (*p* = 0.45). In total, 63% of the patients were operated on laparoscopically, without statistical differences between the two groups (21 versus 11 patients, *p* = 0.11), while the rates of conversion were comparable and low in both group (3.4% vs. 4.5%, *p* = 0.67) (Table 1). The operating times were also not significantly different between the groups (169 min versus 161 min, *p* = 0.53).

### 3.2. Histopathological Analyses

Postoperatively, the histopathological analyses revealed that the inflammatory activity levels of the resected specimens were similar between both groups (*p* = 0.15). The specimens demonstrated predominantly medium or high inflammation activity. Regarding the resection margins, the rates of positive margins for inflammation were also comparable (13 versus 11, *p* = 0.97), without any differences in relation to the region of the positive margin (oral (proximal), aboral (distal), or both) (Table 2).

### 3.3. Short-Term Postoperative Outcome

When analyzing the short-term postoperative outcome of patients following ICR, no differences were seen regarding the length of hospital stay (8.1 versus 8.1 days, *p* = 0.95). Similarly, the postoperative complications detected and evaluated using the Clavien–Dindo classification demonstrated a comparable CCI for both groups (9.7 versus 13.0, *p* = 0.38), as well as similar rates of severe complications (Clavien–Dindo > IIIa) (3 versus 4, *p* = 0.43). In addition, no differences were seen for rates of anastomotic leakage (0 versus 1, *p* = 0.26) and postoperative ileus (3 versus 4, *p* = 0.45), while there was a trend of increased rates of wound infections following Kono-S anastomosis (6 versus 10, *p* = 0.06, Cohen’s *d* = −0.51) (Table 2).

### 3.4. Disease Recurrence

The inflammatory activity was evaluated and described using the Rutgeert score during endoscopy following ICR (time lapse 5–15 months, median 8.8 months) according to international guidelines. The Rutgeert score was detected in all patients and tended to be decreased in patients who received the Kono-S anastomosis in comparison to the conventional side-to-side anastomosis, without reaching statistical significance (1.7 versus 2.5, *p* = 0.11, Cohen’s *d* = 0.47). However, the rates for patients without any endoscopic signs of inflammatory activity (Rutgeert score i0) tended to be higher six to twelve months after the Kono-S reconstruction (17.2 versus 31.8%, *p* = 0.23, Cohen’s *d* = −0.35), whereas patients receiving the conventional side-to-side anastomosis demonstrated increased rates of more severe inflammation (Rutgeert score > i2) (44.8 versus 31.8%, *p* = 0.36, Cohen’s *d* = 0.26).

## 4. Discussion

The implementation of optimal therapeutic regimens for patients suffering from Crohn’s disease is challenging due to the complexity and heterogeneity of the disease, with the rates of surgery remaining relevant despite the introduction of novel antibody-based medications. Since disease recurrence following bowel resection is a major issue, with some patients potentially needing further surgeries, novel surgical strategies with a focus on the creation of an anastomosis might be crucial to decrease rates of clinical and endoscopic disease recurrence [10,18]. By introducing the Kono-S anastomosis, Kono et al. addressed the role of the mesentery on intestinal inflammation by relocating the anastomosis away from it (anti-mesenteric). While the initial data on this modern technique were impressive in selected patients, further evidence about the Kono-S anastomosis in comparison to the conventional side-to-side anastomosis is still lacking, especially in non-selective patient cohorts, despite the great relevance of the issue [8,17]. Therefore, we analyzed the feasibility and safety of the Kono-S anastomosis during its novel implementation at our department and evaluated the rates of disease recurrence. Based on our data of a non-selective patient cohort, the creation of the Kono-S anastomosis following ICR resulted in decreased rates of disease recurrence postoperatively with comparable rates of complications.

After the implementation of the Kono-S anastomosis in our department, the endoscopic recurrence rates as evaluated and described by the Rutgeert score trended to be decreased during postoperative follow-up for patients receiving the Kono-S anastomosis in comparison to those patients treated with the conventional side-to-side anastomosis (1.7 versus 2.5, *p* = 0.11, relevant effect size). With these results being close and not statistically significant, this seemed to be mainly due to the size of the patient cohort. In addition, no patient selection process was performed preoperatively and the follow-up period was limited, which could further affect and explain the moderate statistical differences. However, our results have great clinical relevance, since further calculations revealed that the number of patients without any signs of endoscopic disease recurrence (Rutgeert score i0) increased by 14.6% with the Kono-S anastomosis (17.2 versus 31.8%, *p* = 0.23, medium effect size). Moreover, performing the conventional side-to-side anastomosis resulted in higher rates of severe disease recurrence (Rutgeert score > i2) in comparison to the Kono-S anastomosis (44.8 versus 31.8%, *p* = 0.36, low effect size) (Table 2). Importantly, no differences were observed between the groups in regard to serious postoperative complications, underlining the feasibility and safety of the Kono-S anastomosis during its implementation. In line with this, we identified a learning curve of approximately 20 procedures in our department. While the rates of wound infection tended to be increased for the Kono-S anastomosis technique (6 versus 10, *p* = 0.06, relevant effect size), this might be related to the different access route used to perform the anastomosis (suprapubic for the side-to-side anastomosis versus periumbilical incision for the Kono-S anastomosis). However, no patient selection process was performed in our cohort, as demonstrated by the comparable patient characteristics for their co-morbidity and disease histories, including for previous operations as well as medical treatments (Table 1), which reflects the clinical routine.

Importantly, our study is one of the first studies to investigate the effect of the Kono-S anastomosis technique in patients receiving a re-operation. Based on the results, our analysis supports the further evaluation of the Kono-S anastomosis technique following ICR in patients with Crohn’s disease in clinical routine in non-selective patient cohorts. In line with previous small studies focusing mainly on the perioperative morbidity of the Kono-S anastomosis [19,20,21], the rates of complications in our cohort were comparable between both groups, including the operating time and length of hospital stay (Table 2). Furthermore, while our data demonstrated a clear trend towards decreased rates of disease recurrence, a multicenter study even demonstrated a five-year surgical recurrence-free survival rate of 98.6% in Japan [22]. In addition, another study from Japan also demonstrated decreased rates of anastomotic surgical recurrence following a Kono-S anastomosis after a one-year follow-up [23]. However, no endoscopic disease assessment was performed in this study, which is considered the standard of care for disease monitoring and management in the current guidelines. Following the standardized follow-up of the operated patients, Kono et al. showed significantly lower numbers of patients suffering from disease recurrence in their initial cohort, with a mean Rutgeert score of 0.78 during a follow-up of more than a year [16]. The only small prospective randomized trial (SuPREMe trial), which included 36 patients who received a Kono-S anastomosis, as well as 43 patients who received a conventional side-to-side anastomosis, demonstrated significantly decreased rates of endoscopic recurrence following the creation of the functional end-to-end handsewn anastomosis (Kono-S) [24]. Importantly, and in line with our study, in the SuPREMe trial, the rates of severe endoscopic recurrence (Rutgeert score > i2) were also relevantly lower for patients who received the Kono-S anastomosis, while the numbers of postoperative complications were comparable between both groups. A systematic review by Alshantti et al. confirmed the positive results of the Kono-S anastomosis on postoperative disease recurrence and morbidity [17].

A major aspect and potential explanation for the reduced rates of postoperative recurrence in patients who received the Kono-S anastomosis is the exclusion of the mesentery by the anti-mesenteric creation of the anastomosis. While the role of the mesentery in Crohn’s disease is still controversial [25], Coffey et al. demonstrated that the inclusion of the mesentery in ileocolic resections might further reduce the incidence rates of disease recurrence following surgery [26]. To further address this aspect, future studies such as the SPICY trial will analyze the ongoing discussion about the role of the mesentery in Crohn’s disease [27,28]. Furthermore, another relevant and well-discussed issue is the effect of the positive resection margins on the rates of postoperative disease recurrence [10]. The current evidence is heterogenous on the question of whether a positive resection margin results in a higher risk of disease recurrence [29,30,31]. While the current guidelines do not recommend inflammation-free margins due to the lack of robust evidence and the importance of bowel-sparing resections, future strategies could focus on the relevance of intraoperative diagnostics to avoid positive resection margins, as this is the state-of-the-art approach in surgical oncology. However, while we did not observe differences in resection margins in our cohort between the groups, no conclusion could be drawn from our analysis on the role of positive margins, since the cohort was too small for a subgroup analysis. Randomized studies are necessary with a focus on the mesentery as well as resection margins to further improve and optimize the surgical techniques used in Crohn’s disease.

Our study has several limitations, including its retrospective character, as well as the single-center design. In addition, several patients were lost during the follow-up due to the organization of the German health care system, having a large private practice sector, resulting in a smaller number of included patients, which limited the statistical analyses. However, our group sizes are in line with other published studies on the Kono-S anastomosis technique, as well as surgery in Crohn’s disease in general, since CD is highly heterogenous, including the location of the inflammation. Furthermore, no patient selection process was performed in almost any patients receiving the Kono-S anastomosis following the introduction of it at our department. In addition, we also included patients who had undergone a previous Crohn’s-disease-associated surgery, which explains the primarily open approach in many of our patients (*n* = 17). Therefore, our cohort represents the clinical routine, without any modification.

## 5. Conclusions

In conclusion, we demonstrated in our single-center study that the Kono-S anastomosis technique is feasible and safe during its implementation. While some aspects of our study limit our ability to make a final conclusion about the role of the Kono-S anastomosis in postoperative disease recurrence in non-selected patients, our study supports the need for further investigations of the technique in patients with localized CD. Future randomized trials are necessary to confirm and extend our data and to further improve the surgical strategies to optimize patient care and decrease rates of postoperative disease recurrence.

## Figures and Tables

**Figure 1 jcm-11-06915-f001:**
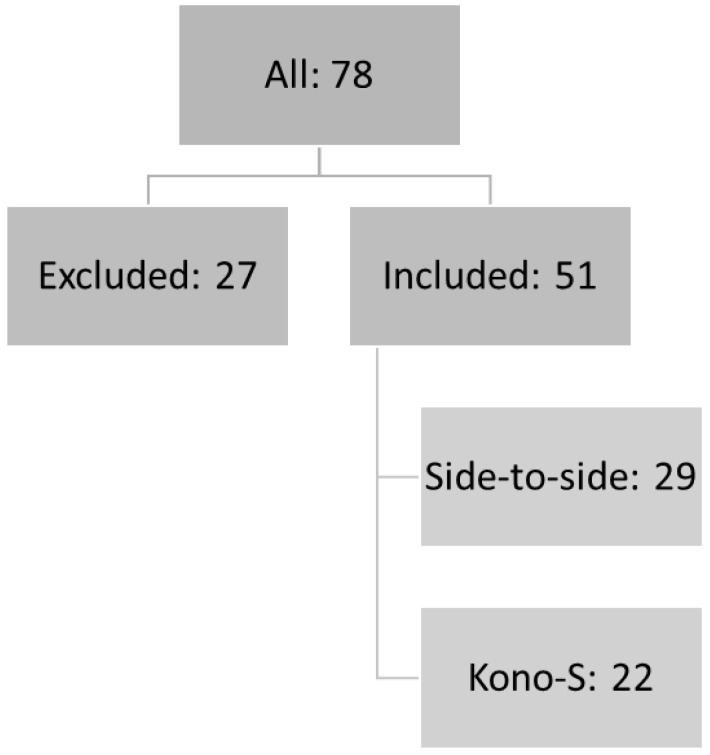
Study design.

**Table 1 jcm-11-06915-t001:** Patient characteristics.

	Side-To-Side (*n* = 29)	Kono-S (*n* = 22)	*p*-Value
Age, mean (years)	36.8 (16–71)	37.4 (20–62)	0.91
Sex (*n*)			0.28
women	15	8	
men	14	14	
BMI, mean (kg/m^2^)	22.8	24.3	0.20
Smoking, *n*	11	5	0.25
Co-orbidities, *n*			
cardiovascular	0	1	
pulmonary	2	2	
diabetes	0	0	
Albumin preoperatively	4.2	4.2	0.72
Hemoglobin preoperatively (g/dL)	12.9	13.3	0.42
Montreal-classification, *n*			0.46
B1	3	1	
B2	22	17	
B3	4	4	
Time from diagnosis to surgery, mean (years)	10.2	7.9	0.46
Previous surgery, *n*			0.59
yes	6	6	
no	23	16	
Immunosuppression, *n*			0.05
none	8	12	
cortisone	9	4	
azathioprine	0	1	
tnfa-inhibitor	7	4	
biological *	5	1	
Type of surgery, *n*			0.31
laparoscopic	21	12	
open	7	10	
Rate of conversion, *n* (%)	1 (3.4)	1 (4.5)	0.67
Operating time, min	168	161	0.53

* All biological therapies excluded TNFa inhibitors.

**Table 2 jcm-11-06915-t002:** Histopathological analysis and postoperative outcomes.

	Side-To-Side (*n* = 29)	Kono-S (*n* = 22)	*p*-Value
Positive resection margin, *n*			0.97
negative	16	12	
oral	8	5	
aboral	2	3	
both	3	2	
Histological inflammatory activity, *n*			0.15
low	1	1	
medium	7	10	
high	21	11	
complications, *n*			
anastomotic leakage	0	1	0.26
ileus	3	4	0.45
wound infection	6	10	0.06
CCI, median	9.7	13.0	0.38
Clavien-Dindo > IIIa	3	4	0.43
Postoperative immunosuppression, *n*			0.45
none	15	13	
cortisone	1	0	
azathioprine	0	1	
tnfa-inhibitor	7	7	
biological *	6	1	
Length of hospital stay, days	8.1	8.1	0.95
Readmission, *n*	0	2	0.10
Rutgeert score postoperatively, mean	2.5	1.7	0.11
Rutgeert score (%)			
i0	5 (17.2)	7 (31.8)	0.23
>i2	13 (44.8)	7 (31.8)	0.36

* All biological therapies excluded TNFa-inhibitors.

## Data Availability

The data presented in this study are available on request from the corresponding author.

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
