# Peer review of "Kono-S Anastomosis in Crohn’s Disease: A Retrospective Study on Postoperative Morbidity and Disease Recurrence in Comparison to the Conventional Side-To-Side Anastomosis"

_jcm, 2022, doi:10.3390/jcm11236915_

Round 1

Reviewer 1 Report

 The authors investigated the feasibility and safety of the Kono-S anastomosis for patients with Crohn’s disease compared with conventional side-to-side anastomosis. Although they described the trend towards decreased rate of recurrence in patients with Kono-S anastomosis, there was no significant difference. They failed to prove their hypothesis in this report, probably owing to the small sample size. In addition, the findings which they described in this report are not novel. As they described, the prospective randomized trial (SuPREMe-trial), which included a little larger cohort than that of this retrospective report, had demonstrated significantly decrease rate of recurrence in patients with Kono-S anastomosis. I am not certain what is substantively new in this manuscript.

Author Response

Dear Dr. Wang,
Dear Reviewers,
Thank you for giving us the opportunity to reply to your comments on our manuscript entitled “Kono-S anastomosis in Crohn`s Disease: a retrospective study on postoperative morbidity and disease recurrence in comparison to the conventional side-to-side anastomosis”.
We appreciate the time and effort that you and the reviewers have dedicated to providing us your valuable feedback on our manuscript. We are grateful to the reviewers for their insightful comments on this paper. Please see below a point-by-point response to the reviewers’ comments and concerns. All changes are highlighted using the “Track Changes” function in the manuscript.

Comments from Reviewer 1:
The authors investigated the feasibility and safety of the Kono-S anastomosis for patients with Crohn’s disease compared with conventional side-to-side anastomosis. Although they described the trend towards decreased rate of recurrence in patients with Kono-S anastomosis, there was no significant difference. They failed to prove their hypothesis in this report, probably owing to the small sample size. In addition, the findings which they described in this report are not novel. As they described, the prospective randomized trial (SuPREMetrial), which included a little larger cohort than that of this retrospective report, had demonstrated significantly decrease rate of recurrence in patients with Kono-S anastomosis. I am not certain what is substantively new in this manuscript.

Answer: We really would like to thank the reviewer for this important and critical remark. One of the aims of our study was to show that even in the period of clinical implementation and during the learning curve the Kono-S anastomosis is a safe technique without increased perioperative morbidity and mortality. In addition, our data demonstrate that this new surgical technique also results in potential long-term benefits for patients suffering from complicated ileitis terminalis Crohn who are not monitored under study conditions such as in the SuPREMe trial. Importantly, in the SuPREMe trial only a selective patient cohort was included which does not represent the daily routine of a highly heterogenous disease such as CD. Moreover, we think that our results would have shown also significant results regarding the Rutgeert score if we would have increased the number of included patients. However, since we also focused on a potential learning curve and identified roundabout 20 procedures as cut off based on our experience, we did not want to enlarge the study population.
Furthermore, it must be considered that our study covers the time of COVID-19 pandemic with its limitation of elective operations. Despite those limitations, our study still demonstrates relevant differences between both anastomosis techniques since rates of postoperative recurrence are decreased following Kono-S. Only because it is not statistical significantly different, it does not mean it is not clinically relevant. Therefore, due to the limited evidence available with only one prospective randomized trial, we strongly believe that there is still a need for further evidence demonstrating that this surgical technique is safe and provides benefits for Crohn patients outside randomized and controlled studies during daily routine, even in the period of learning curve.

Reviewer 2 Report

Retrospective cohort study comparing post op recurrence for CD patients undergoing conventional side-to-side anastomosis vs Kono-S anastomosis after ileocecal resection.

The topic is interesting and clinically relevant. However the study in its current form is not able to provide any clear statement on the post op recurrence between those two techniques. My main concern is that the cohort is underpowered. Have the authors conducted a power conculation prior to analysis? if they did, please add to the statistical methods.

My second major concern is the disconnection between the results and the conclusion and the elimination of the p values from the abstrast. The analysis did not demonstrate any significant differences between outcomes in each group. Trend does not mean statistical significant. However, the non-significant p vlaues are not included in the abstrast and the conclusion is not backed by the results. This must be corrected and the conclusion must be toned down accordingly.

Minor - 

line 36-37 - please tone down the statement that new guidelines  recommend surgical induction, it is still not the common practice in newly diagnosed inflammatory phenotype (Lyric type patients). Indeed, in the cohort included in this study patients were about 10 years from diagnosis and most had stricturing phenotype. Please modify.

lines 48-49, 51 - add the relevant references

Table 1 - 1. add % throughout the entire table, 2. correct therapies section - there are typos in medication names (cortisone, azathiporine). What is the different between anti-TNF and biologics (which include anti-TNF), if including non-TNF please modify accordingly. 

line 124 - how inflammatory activity in the resected segment was evaluated? this is not described at all in the methods section. Please clarify and add.

line 128 and table 2 - I asssume that oral and aboral suppose to be normal and abnormal. Please correct. 

English should be improved throughout the manuscript 

Rutgeert-Score 1.7 versus 2.5) with relevantly increased rates of patients 

with complete disease remission (17.2% versus 31.8%).

Conclusion: Kono-S anastomosis is safe and feasible and potentially decreases 

the severity of postoperative disease remission

Author Response

Dear Dr. Wang,
Dear Reviewers,
Thank you for giving us the opportunity to reply to your comments on our manuscript entitled “Kono-S anastomosis in Crohn`s Disease: a retrospective study on postoperative morbidity and disease recurrence in comparison to the conventional side-to-side anastomosis”.
We appreciate the time and effort that you and the reviewers have dedicated to providing us your valuable feedback on our manuscript. We are grateful to the reviewers for their insightful comments on this paper. Please see below a point-by-point response to the reviewers’ comments and concerns. All changes are highlighted using the “Track Changes” function in the manuscript.

Comments from Reviewer 2:
1. Retrospective cohort study comparing post op recurrence for CD patients undergoing conventional side-to-side anastomosis vs Kono-S anastomosis after ileocecal resection. The topic is interesting and clinically relevant. However, the study in its current form is not able to provide any clear statement on the post op recurrence between those two techniques. My main concern is that the cohort is underpowered. Have the authors conducted a power conculation prior to analysis? if they did, please add to the statistical methods.

Answer: We thank the reviewer for this comment. We agree that the topic is clinically relevant and important but still not finally answered if this new technique is superior to conventional side-to-side anastomosis. Thus, further evidence is mandatory, not only from further randomized prospective trials with larger study population as in the SuPREMe trial but also real-world data as provided from our study. As we mentioned already above, our aim was also to show that the Kono-S anastomosis is a safe surgical procedure even in times of clinical implementation and learning curve. Therefore, we did not conduct a power calculation.

2. My second major concern is the disconnection between the results and the conclusion and the elimination of the p values from the abstrast. The analysis did not demonstrate any significant differences between outcomes in each group. Trend does not mean statistical significant. However, the non-significant p vlaues are not included in the abstrast and the conclusion is not backed by the results. This must be corrected and the conclusion must be toned down accordingly.

Answer: We thank the reviewer for these remarks. We included the p-values and toned down our conclusion (Line 243-248). However, in the first draft, we already mentioned that the Kono-S anastomosis might decreased the postoperative recurrence (line 242) and Rutgeert-Score trended to be decreased during postoperative follow-up (line 168-169).

3. Line 36-37 – please tone down the statement that new guidelines recommend surgical induction, it is still not the common practice in newly diagnosed inflammatory phenotype (Lyric type patients). Indeed, in the cohort included in this study patients were about 10 years from diagnosis and most had 2tructuring phenotype. Please modify.

Answer: We agree that the early surgical therapy of isolated ileitis terminalis is still not the common practice but guidelines a such as the new German Crohn´s guideline or ECCO guideline recommend that surgical therapy should be discussed with patients before an antibody-based therapy is started. This recommendation is based on the results from the LIR!C trial as mentioned in the manuscript. As asked by the rewiever, we modified the statement and changed “recommend” to “consider” (Line 36). Our study group has been recently published a narrative review about the early surgical therapy of patients suffering from isolated ileitis terminalis where we summarized the actual international recommendations regarding this disease type (Kelm M, Germer CT, Schlegel N, Flemming S. The Revival of Surgery in Crohn's Disease-Early Intestinal Resection as a Reasonable Alternative in Localized Ileitis. Biomedicines. 2021 Sep 26;9(10):1317. doi: 10.3390/biomedicines9101317. PMID: 34680434; PMCID: PMC8533348.) Please see the table below(attachment).

4. Lines 48-49, 51 - add the relevant references

Answer: We added proper references as asked by the reviewer (Line 48-50).

5. Table 1 - 1. add % throughout the entire table, 2. correct therapies section - there are typos in medication names (cortisone, azathiporine). What is the different between anti-TNF and biologics (which include anti-TNF), if including non-TNF please modify accordingly.

Answer: We thank the reviewer for this advice. Numbers presented in Table 1 are mainly total numbers and not percentages. Other than that, we corrected it accordingly your suggestion in the manuscript. We are sorry for this misunderstanding. Indeed, biologicals includes all biological therapies without TNFa-inhibitors which we listed separately.

6. Line 124 - how inflammatory activity in the resected segment was evaluated? this is not described at all in the methods section. Please clarify and add.

Answer: Inflammatory activity was evaluated based on histopathological investigations (mucosal damage, immune cell infiltration) and was divided into three categories (low, medium, high). We mentioned this already in the manuscript but extended our explanation (Line 84-87).

7. Line 128 and table 2 - I asssume that oral and aboral suppose to be normal and abnormal. Please correct.

Answer: With oral and aboral margin we described the proximal and distal resection margin. We have clarified that issue in the manuscript.

Reviewer 3 Report

In the present study, the authors investigate short and long-term outcomes of Kono-S versus side to side anastomosis at a single center. They found that the two techniques have comparable short term outcomes, and trend towards reduced disease recurrence postoperatively. 

A few questions and comments are shared below:

1. The introduction is excellent - sets the stage nicely for the study. 

2. The limited sample size is a weakness of this study and I believe precluded some of the results from reaching significance. On a related note, although there are largely no significant differences between baseline characteristics, the rates of preoperative immunosuppression trended to be higher for patients who received the side to side anastomosis - could this be reflective of more severe baseline disease that in turn lead to the trend towards higher disease recurrence postoperatively?

3. While the study question is of interest, my biggest concern with this paper is the novelty (or lack thereof). There seem to be other studies performed with larger sample sizes and similar methodologies (i.e. with post-op endoscopic evaluation) that reached similar conclusions (and with statistical significance given the larger sample sizes). Thus, I would challenge the authors to modify their study design in such a way that the results add to the existing literature (instead of merely supporting it with sub-optimal sample sizes), or perhaps frame their current findings in a way that add novelty.

Reviewer 4 Report

Thank you for the possibility to review the article entitled “Kono-S anastomosis in Crohn`s Disease: a retrospective study on postoperative morbidity and disease recurrence in comparison to the conventional side-to-side anastomosis”. Overall, the article is well conducted and written. I have few remarks.

Please uniform the number of decimals within the text and tables. Moreover, numbers should not be used at the beginning of the sentence (e.g., Twenty-six rather than 26).

My only consistent concern is regarding the interpretation of your results.  You stated that the rate of postop complications, such as infection was similar in the two groups because you did not find a p<0.05. On the other hand, (see line 170) you stated that the rate of recurrence was inferior in the Kono-S group even a p<0.05 was not found. If you do so, the number of reoperations should be also considered higher in the Kono-S group. I suggest also to consider the effect size, not just p values. Considering the relatively small sample size it might be even more informative than the p values. I would consider the rate of wound infection very significant (big effect size, p NS), while the Rutgeert-Score is less significant (low effect size, p NS). The interpretation of the Rutgeert-Score i0 groups is correct (relevant effect size although p NS).

Finally, you mentioned that the patients underwent an endoscopy between 6-12 months postoperatively. However, the timelapse is missing and should be reported as it may affect the Rutgeert-Score.

Round 2

Reviewer 1 Report

 According to the authors’ answer, the novelty of their study is the non-selective, routinely treated patients’ cohort, which reflects the disease heterogeneity. If it is one of their aims, they should provide more detail discussion about it, including whether the non-selective cohort might affect the non-significant results or not. Of cause, lack of statistically significance does not mean “not relevant”. It only means that you can not say there is a difference. They should carefully discuss the reasons: sample size, case selection, follow-up period etc.

 In addition, they mentioned that they focused on a potential learning curve. As far I understand, the manuscript did not show about learning curve. If they focused on a learning curve in this study, the meaning of this study is quite different from that of general surgery trials. They should appropriately describe it in the manuscript.

Author Response

According to the authors’ answer, the novelty of their study is the non-selective, routinely treated patients’ cohort, which reflects the disease heterogeneity. If it is one of their aims, they should provide more detail discussion about it, including whether the non-selective cohort might affect the non-significant results or not.
Of cause, lack of statistically significance does not mean “not relevant”. It only means that you can not say there is a difference. They should carefully discuss the reasons: sample size, case selection, follow-up period etc.

In addition, they mentioned that they focused on a potential learning curve. As far I understand, the manuscript did not show about learning curve. If they focused on a learning curve in this study, the meaning of this study is quite different from that of general surgery trials. They should appropriately describe it in the manuscript.

Answer: We thank the reviewer for the comment. While it was already mentioned before in Line 76-79 that no patient selection was performed and almost all patients received the Kono-S anastomosis after its introduction, we agree that both aspects listed above were not discussed in detail in the manuscript and, therefore, edited the manuscript accordingly (Line 56-57, 61, 81-82, 168-171, 179-181, 188-190, 197-199, 253). It is correct that we mainly focused on the feasibility and safety of the technique during its implementation as well as the postoperative outcome for Kono-S on disease recurrence in a non-selective cohort, therefore, identifying a potential learning curve was only a secondary aspect (Line 81-82).

Reviewer 2 Report

I respectfully disagree with the authors since that analysis, results or conclusion, did not include only safety data in real world setting but clinical disease outcomes as well. Therefore, with not power analysis and in light of no significant results no conclusion can be made. If there is previous data to compare the two techniques power analysis can be done based on the previous reported data. The current conclusion regarding a possible advantage in not more than a wishful thinking and cannot be made based on the current analysis without power calculation.

Author Response

I respectfully disagree with the authors since that analysis, results or conclusion, did not include only safety data in real world setting but clinical disease outcomes as well. Therefore, with not power analysis and in light of no significant results no conclusion can be made. If there is previous data to compare the two techniques power analysis can be done based on the previous reported data. The current conclusion regarding a possible advantage in not more than a wishful thinking and cannot be made based on the current analysis without power calculation.

Answer: We agree with the reviewer that a power analysis would enhance the conclusion of the study.
However, as we mentioned before one goal of the study was to evaluate the technique during its implementation and in a non-selective cohort, therefore, increasing the cohort following a power analysis would undermine the aspect of evaluating the feasibility while introducing a novel technique since the amount of patient to included would probably increase. We also agree that the conclusion regarding disease recurrence is limited without a power analysis, however, since our study also focuses on several other aspects including feasibilty/safetiness during implementation, non-selective cohort, learning aspect, we do think that our manuscript provides important information on the highly relevant question of anastomotic techniques in CD patients. To better address the reviewer`s aspect, we rephrased again our conclusion (Line 253-256).

Reviewer 3 Report

Dear authors,

Thank you for taking the time to respond to my comments. You mention that one of the unique aspects of the study is to demonstrate that Kono-S is safe even as a newly implemented technique, which may be of interest to centers who hesitate to adopt it due to fear of increased complications during the early adoption period. I note that you have revised your manuscript to further emphasize this point. The other points you bring up are also important; I believe you mentioned some of them in your discussion already, but would make sure these are clearly delineated to once again emphasize what this particular study contributes to existing literature. 

As long as the aforementioned points are clearly described (as well as the limitations already discussed (i.e. sample size)), the study can be framed in a slightly different perspective to make it clinically relevant.

Of note, in line 167 - please change "safetiness" to "safety"

Author Response

We thank the reviewer for the positive comments. After critical revision of our manuscript, we do think that our main points are described accordingly to underline the relevance of the study. However, to address some aspects better, we again edited the discussion in some parts as recommended by the reviewer (Line 169, 200-201).

Round 3

Reviewer 2 Report

The authors must rephrase the abstract to reflect the lack of significance and to avoid missleading of the readers: "Disease recurrence postoperatively was numerically lower following Kono-S anastomosis (median Rutgeert- 23 Score 1.7 versus 2.5) with relevantly increased rates of patients in remission 24 (17.2% versus 31.8%), however, none of which reached statiscial significance. "

Author Response

We thank the reviewer for the observation and corrected the sentence according to the suggestion